# Polar confinement of a macromolecular machine by an SRP-type GTPase

Anita Dornes [1,5], Lisa Marie Schmidt[2,5], Christopher-Nils Mais[1], John C. Hook[2], Jan Pané-Farré [1], Dieter Kressler [3], Kai Thormann [2] ✉ & Gert Bange [1,4] ✉

The basal structure of the bacterial flagellum includes a membrane embedded MS-ring (formed by multiple copies of FliF) and a cytoplasmic C-ring (composed of proteins FliG, FliM and FliN). The SRP-type GTPase FlhF is required for directing the initial flagellar protein FliF to the cell pole, but the mechanisms are unclear. Here, we show that FlhF anchors developing flagellar structures to the polar landmark protein HubP/FimV, thereby restricting their formation to the cell pole. Specifically, the GTPase domain of FlhF interacts with HubP, while a structured domain at the N-terminus of FlhF binds to FliG. FlhF-bound FliG subsequently engages with the MS-ring protein FliF. Thus, the interaction of FlhF with HubP and FliG recruits a FliF-FliG complex to the cell pole. In addition, the modulation of FlhF activity by the MinD-type ATPase FlhG controls the interaction of FliG with FliM-FliN, thereby regulating the progression of flagellar assembly at the pole.

The flagellum is a macromolecular machine, which enables the movement of bacteria along chemical gradients[1]. The core flagellar architecture is conserved, and it is composed of the MS-ring, cytoplasmic C-ring, the rod, and extracellular hook and filament (Fig. 1a). The membrane-embedded MS-ring is formed by multiple copies of a single transmembrane protein FliF[2–5]. At the cytoplasmic side of the MS-ring resides the flagellar C-ring, an oligomeric structure of the proteins FliG, FliM and FliN[6,7], and required for power transmission, in both counter-clockwise and clockwise rotational modes of the flagellum.

The number and arrangement of flagella give rise to unique "flagellation patterns," which vary between bacterial species but remain characteristic to each[8–10]. However, the molecular mechanisms controlling the spatial-numerical distribution of flagella are still far from being understood. The FlhF protein, in conjunction with the MinD-type ATPase FlhG, also referred to as YlxH, FleN, MotR, or MinD2, plays a crucial role in determining the positioning and assembly of flagella in numerous polar and peritrichous flagellated bacteria (reviewed in refs. 8,9:). FlhF is essential for directing the initial flagellar protein, FliF, to

the cell pole, although the exact mechanism remains incompletely understood[11,12].

FlhF belongs to the family of signal recognition particle (SRP)-GTPases, and shares its NG domain with the other two members of the family (i.e., Ffh and FtsY)[13,14]. The GTPase activity of FlhF is stimulated by FlhG, through a conserved "DQAxxLR" motif present at its N-terminus[15–17]. In contrast to the other two SRP-GTPases, FlhF possesses an N-terminal B-domain believed to be structurally disordered and implicated in the targeting of FliF (Fig. 1b[11,12];). In addition, we recently identified a FlhF-interacting protein, named FipA, which, in concert with the polar landmark protein HubP/FimV, is involved in targeting FlhF to the cell pole in *Vibrio parahaemolyticus*, *Pseudomonas putida* or *Shewanella putrefaciens*[18]. However, the molecular mechanism by which FlhF enables assembly of the flagellum at one cell pole in monotrichous bacteria is still elusive.

Thus, we set out to shed light on the molecular mechanism enabling FlhF to position the flagellum in polar flagellates. To this end, we used *S. putrefaciens* CN32 as our model system, in which we have previously studied in detail the flagellar regulation, mechanism and

¹Philipps-University Marburg, Center for Synthetic Microbiology (SYNMIKRO) and Department of Chemistry, Hans-Meerwein-Strasse 6, C07, 35043 Marburg, Germany. ²Justus-Liebig-Universität, Department of Microbiology and Molecular Biology, Heinrich-Buff-Ring 26, 35392 Giessen, Germany. ³University of Fribourg, Department of Biology, Chemin du Musée 10, 1700 Fribourg, Switzerland. ⁴Max-Planck-Institute for terrestrial Microbiology, Molecular Physiology of Microbes, Karl-von-Frisch Strasse 14, 35043 Marburg, Germany. ⁵These authors contributed equally: Anita Dornes, Lisa Marie Schmidt. ✉e-mail: kai.thormann@mikro.bio.uni-giessen.de; gert.bange@synmikro.uni-marburg.de

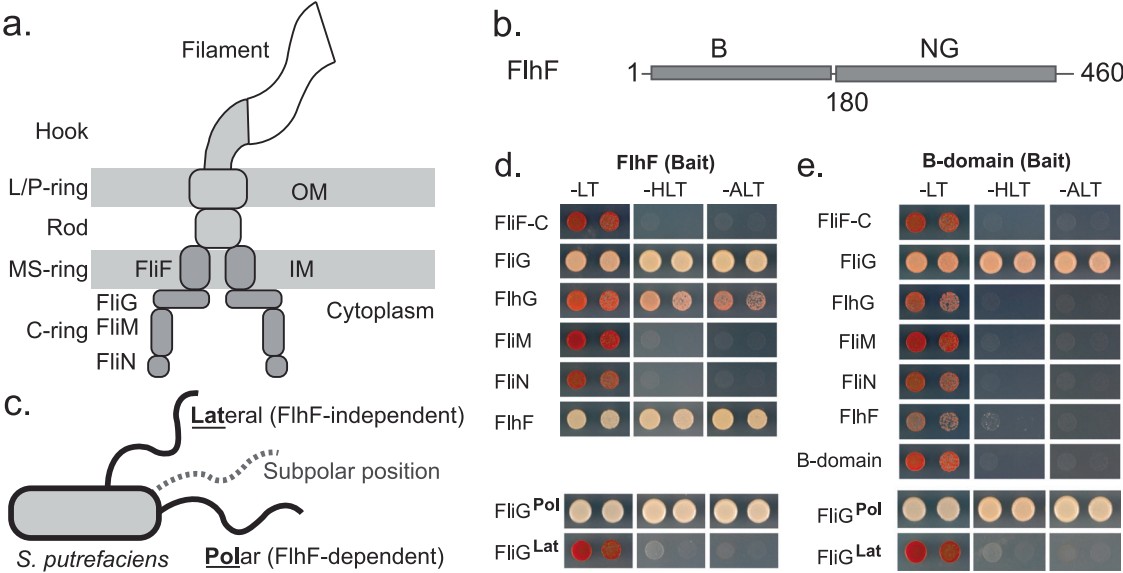

**Fig. 1 | Essential role of FlhF's B-domain in interacting with FliG.** a Scheme of the architecture of the bacterial flagellum. **b** Schematic representation of *S. putrefaciens* FlhF's domain structure. **c** Scheme of the flagellation of the Gram-negative model organism *S. putrefaciens* featuring one polar and one lateral flagellum, whose localization is dependent and independent of FlhF, respectively. The subpolar position of the flagellum is indicated in dashed gray. **d** *Upper panel*: FlhF interacts with FliG in the yeast two-hybrid (Y2H) assay, while showing no interaction with FliF-C, FliM, and FliN from the polar system. *Lower panel*: FlhF exclusively associates with FliG from the polar flagellar system but not with FliG from the lateral system.

**e** *Upper panel*: The B-domain of FlhF is sufficient to interact with FliG, while no interaction is observed with FliF-C, FliM, and FliN of the polar system. *Lower panel*: The B-domain of FlhF demonstrates its ability to differentiate between the FliG proteins of the polar and lateral systems in the Y2H. For Fig. 1d, e: The growth of cells, co-expressing the FlhF and FlhF-B bait proteins and the indicated prey proteins, was assessed on SC-Leu-Trp (-LT), SC-His-Leu-Trp (-HLT; *HIS3* reporter) and SC-Ade-Leu-Trp (-ALT; *ADE2* reporter) plates. The reporter strain PJ69-4A was used in which the *HIS3* and *ADE2* reporter genes are under the transcriptional control of the GAL1 and GAL2 promoter, respectively.

function of FlhG[19–21]. *S. putrefaciens* harbors two distinct flagellar systems[22,23] (Fig. 1c). The primary main monopolar system depends on FlhF and FlhG, while the secondary lateral system is not affected by these two proteins[16,19]. Our data show that FlhF initiates polar flagellar assembly by facilitating assembly of the flagellar MS-ring by directing a key protein of the cytoplasmic C-ring, FliG, to the designated position.

## Results

### The B-domain of FlhF interacts with FliG in the polar flagellar system, excluding the lateral system

FlhF has been suggested to guide the MS-ring protein, FliF, toward the cell pole[11,12], albeit a molecular mechanism remains elusive. These studies suggested to us that FlhF might execute its function in the context of the MS- and/or C-ring proteins FliF, FliG, FliM and FliN (Fig. 1a).

Thus, we started by conducting a yeast two-hybrid (Y2H) assay using the *S. putrefaciens* proteins with FlhF as the bait protein and FliF, FliN, FliM, or FliG as the prey proteins. Since FliF is a membrane protein, we employed its cytoplasmic domain (FliF-C). The results showed that while FlhF did not interact with FliF-C, FliM, or FliN, it exhibited a strong interaction with FliG (Fig. 1d). To validate this discovery, we also assessed the interaction between FlhF and FliG from the lateral flagellar system (FliG-Lat). Contrary to the robust interaction between FlhF and FliG from the polar system, no Y2H interaction could be observed between FlhF and FliG-Lat (Fig. 1d). Consequently, we conclude that FlhF specifically interacts with FliG from the polar flagellar system in *S. putrefaciens*, while not engaging with the FliG protein lateral flagellar system.

Earlier experiments have indicated that the B-domain of FlhF plays a critical role in the polar targeting of FlhF[11,12]. Consequently, we conducted a Y2H analysis to determine whether the B-domain could interact with FliF-C, FliM, FliN, or FliG. Our results clearly demonstrate that the B-domain is both necessary and sufficient for the interaction between FlhF and the C-ring protein FliG (Fig. 1e). Notably, similar to

the full-length FlhF protein, the B-domain exhibits selectivity, distinguishing between FliG proteins of the polar and lateral flagellar systems (Fig. 1e).

### A structured domain at the N-terminus of FlhF mediates the FliG interaction

To consolidate the interaction of the FlhF B-domain with FliG at the biochemical level, we recombinantly produced a StrepII-tagged B-domain together with FliG in *Escherichia coli* BL21(DE3) and performed a pulldown from the cleared cell lysates. The experiment shows a stoichiometric interaction between FlhF-B and FliG (Fig. 2a, *first lane*). In the next step, we performed the same experiment probing the ability of different B-domain truncations to interact with FliG. Only when the first 60 amino acids of the B-domain were fully present, an interaction with FliG could be observed (Fig. 2a, *second lane*). These data show that the N-terminal 60 amino acids are necessary and sufficient for the interaction of FlhF and FliG (Fig. 2a).

Structural analysis by X-ray crystallography of a FlhF construct encompassing the first 44 amino acid residues containing a C-terminal His$_6$-tag for purification, showed that these residues of the B-domain form a domain consisting of three anti-parallelly arranged β-strands and one α-helix (Fig. 2b, Supplementary Table 1, Supplementary Fig. 1a). These data show that the N-terminus of the B-domain, which provides the FliG-interaction site of FlhF is structured. Due to its adept interaction with FliG, we propose labeling this domain as the FliG Interaction Domain (FID).

We also wanted to gain a better understanding of which part of FliG would be required for the interaction with FlhF. Structural analysis showed that FliG consists of three domains, the N-terminal (FliG-N), a middle (FliG-M), and a C-terminal domain (FliG-C). As each of the three domains alone is not stable at the biochemical level, we decided to employ two FliG variants containing either the N- and M-domains (FliG-NM) or the M- and C-domains (FliG-MC) (Fig. 2c, upper panel). Again, we performed pulldown assays with a StrepII-tagged FlhF as bait and

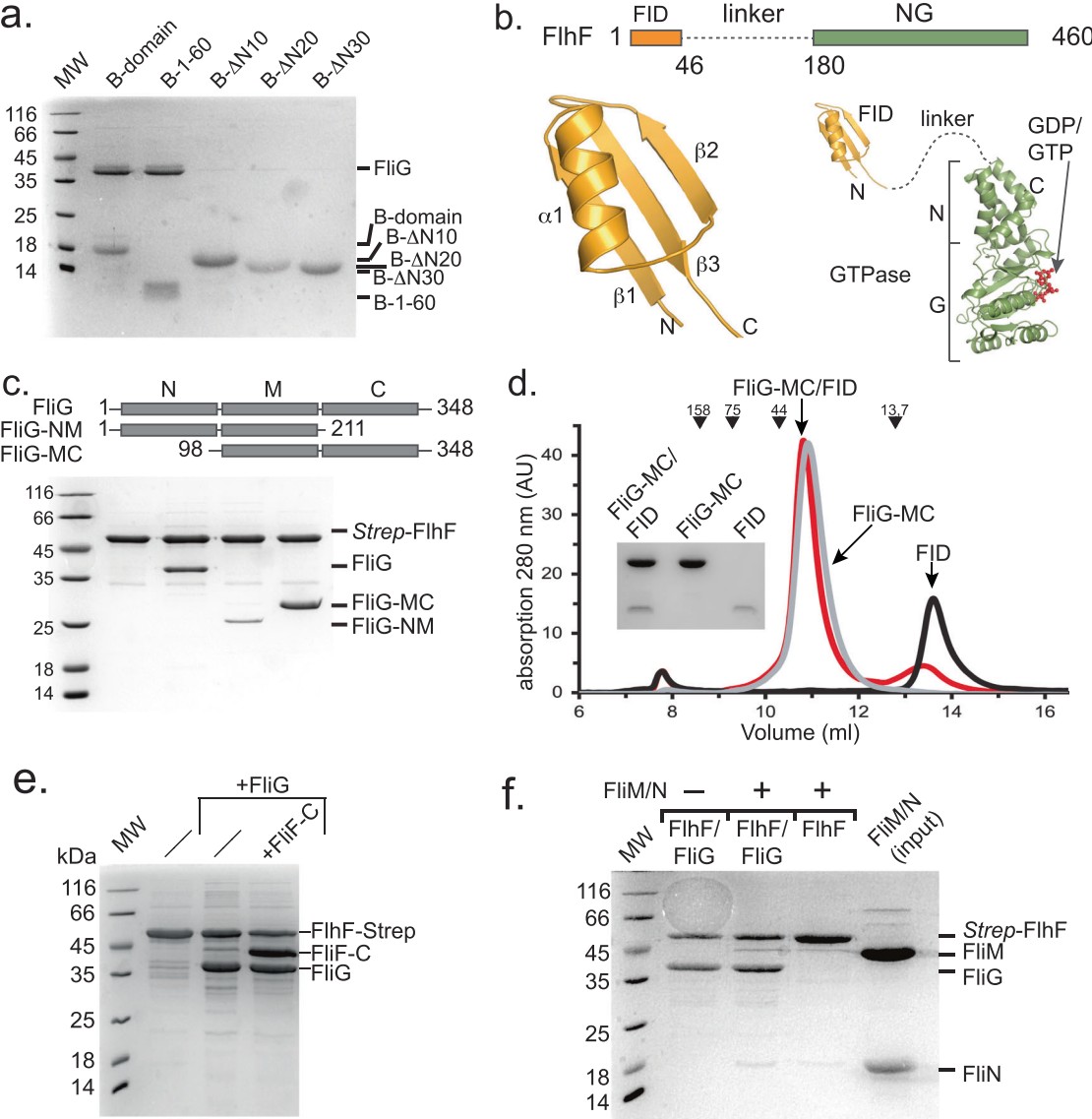

**Fig. 2 | Mechanistic dissection of the FlhF/FliG interaction. a** Coomassie-stained SDS-PAGE of an in vitro pulldown assay employing different StrepII-tagged variants of the B-domain as bait (the amino acid number are indicated) and full-length FliG as prey. **b** Structural analysis of the FID domain of FlhF. *Upper panel*: Revised scheme of the domain architecture of FlhF with the FliG-interacting domain (FID, orange), the structurally uncharacterized linker region (dashed line), followed by the NG domain (green). The domains are drawn to scale. *Lower panel, left*: X-ray structure of the FID domain of FlhF. *Lower panel, right*: X-ray structures of the FID domain (this study) and the GDP-bound state of the NG domain (PDB-ID: 8R9R[49];) from *S. putrefaciens* FlhF. The structurally uncharacterized linker is indicated by a dashed line, not drawn to scale. **c** Coomassie-stained SDS-PAGE employing StrepII-

tagged FlhF as bait and FliG and its variants (given in the panel above) as prey. **d** Chromatogram of an analytical size-exclusion chromatography of the FID domain of FlhF (black), the MC-domains of FliG (grey) and their complex (red). Coomassie-stained SDS-PAGE of the peak fraction of each run is shown in the inset. The arrows show the elution volumes of molecular weight markers (kDa). **e** FlhF-tethered FliG can interact with FliF-C. Coomassie-stained SDS-PAGE employing StrepII-tagged FlhF as bait and FliG or FliG and FliF-C as prey. **f** FlhF-tethered FliG cannot interact with FliM/N. Coomassie-stained SDS-PAGE employing FliG-bound to StrepII-tagged FlhF as bait in the absence and presence of FliM/N. For Fig. 2a, c–f: Each of the shown experiments were repeated in at least three biological replicates. The molecular weight marker (MW) is in kiloDalton (kDa).

FliG, FliG-NM or FliG-MC as prey. FlhF shows a stoichiometric interaction with FliG and FliG-MC, however, its interaction with FliG-NM appeared sub-stoichiometric (Fig. 2c, lower panel). These data strongly suggest that the interaction site of FlhF resides within the M- and C-domains of FliG. Analytical size-exclusion chromatography confirmed the interaction between FlhF-B and FliG-MC (Fig. 2d).

As the next step, our aim was to gain a deeper understanding of the consequences stemming from the FlhF/FliG interaction. Specifically, we sought insights into two aspects: firstly, its impact on the interaction between FliG and the MS-ring protein FliF, and secondly, its influence on FliG's ability to engage with its C-ring counterparts, FliM and FliN. When bound to FlhF, FliG was able to interact with the

cytoplasmic domain of the flagellar MS-ring forming protein FliF (Fig. 2e). Nevertheless, upon binding to FlhF, FliG exhibited an inability to interact with FliM/N (Fig. 2f). These observations underscore that FlhF acts as an impediment, hindering the interaction between FliG and its C-ring partners FliM/FliN, while allowing engagement of FliG to FliF.

## FlhF-GTPase interacts with the cytoplasmic region of HubP
The FlhF-FID interaction with FliG raises the question in which functional context FlhF operates at the cell pole. Previous studies have suggested that FlhF can interact with the polar landmark protein HubP, a hub for various protein interactions[16,24,25]. However, a deeper molecular picture is elusive. HubP is a transmembrane protein with an

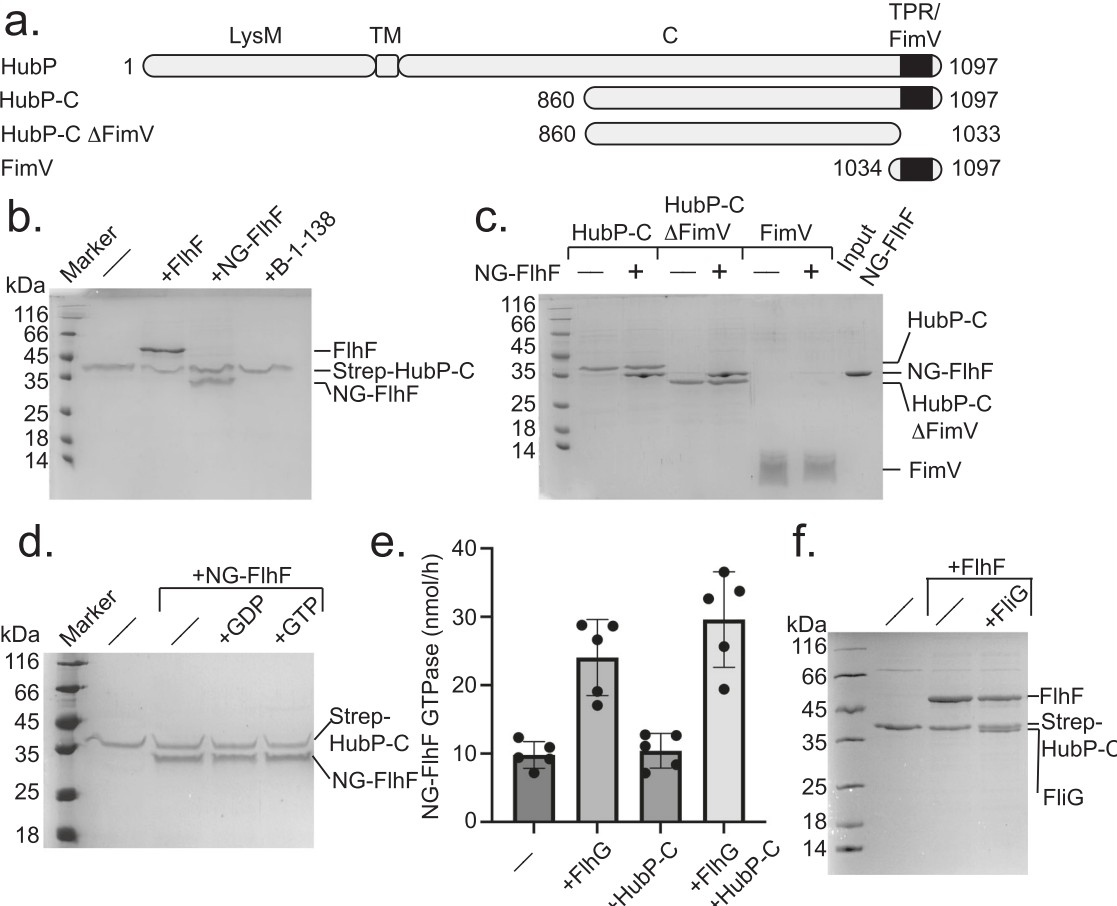

**Fig. 3 | The NG domain of FlhF interacts with the cytoplasmic region of HubP.**
**a** Domain structure of the polar landmark protein HubP/FimV from *S. putrefaciens* and constructs used in this study. **b** Coomassie-stained SDS-PAGE employing StrepII-tagged HubP-C as bait and FlhF, its NG domain or the B-domain as prey. **c** Coomassie-stained SDS-PAGE employing StrepII-tagged HubP-C, StrepII-tagged HubP-C lacking the C-terminal FimV domain or a StrepII-tagged FimV domain alone as bait and NG-FlhF as prey. **d** Coomassie-stained SDS-PAGE probing the impact of GDP or GTP on interaction of StrepII-tagged HubP-C and NG-FlhF, acting as bait and prey, respectively. **e** GTPase activity of NG-FlhF in the presence of its stimulator FlhG, HubP-C and the two together. The bars are mean values from 5 independent experiments, each shown as dot, with the errors shown as standard deviation from the 5 independent experiments. **f** Coomassie-stained SDS-PAGE employing StrepII-tagged HubP-C as bait and FlhF alone and FlhF and FliG. For Fig. 3b, c, d, f: Each of the shown experiments were repeated in at least three biological replicates.

N-terminal LysM-type domain, followed by a transmembrane segment and an extended cytoplasmic region of approximately 70 kDa (HubP-C) (Fig. 3a). In *V. cholerae* and *S. putrefaciens*, HubP localizes as a large cluster to the flagellated cell pole and a smaller minor cluster to the opposite pole. The latter cluster is already formed during cell division in the division plane and starts growing once the cells enter another division cycle[16,26]. How polar identity is maintained between the two clusters remains elusive so far. An Alpha2-fold prediction of HubP including HubP-C is widely unsatisfactory (Supplementary Figs. 1b, c), and predicts the presence of a TPR-repeat, which has been structurally determined for the *Pseudomonas aeruginosa* HubP/FimV[27].

To investigate whether FlhF would interact with the cytoplasmic region of HubP (HubP-C), we designed several HubP variants. However, we could only produce variants starting from amino acid 860 to the C-terminus of the protein. For the pulldowns, FlhF as prey and different StrepII-tagged version of these HubP-C variants were used as baits (Fig. 3a). This experiment shows that FlhF interacts with HubP-C, in a region involving residues 860–1033, not including the C-terminal TPR domain (Fig. 3b). We also probed which of the FlhF domains would be necessary for the FlhF-HubP interaction. We show that the NG domain of FlhF is required for the interaction with HubP-C, while the B-domain is not (Fig. 3c).

Next, we probed whether the interaction of NG-FlhF and HubP-C would depend on the presence of nucleotides. Therefore, we performed in vitro pulldown assays probing whether the addition of GDP or GTP would affect the interaction of NG-FlhF with an StrepII-tagged HubP-C variant immobilized on beads (Fig. 3d). This experiment shows that neither GDP nor GTP affect the NG-FlhF/HubP-C interaction. This notion is supported by GTP hydrolysis assays showing that HubP-C does not affect the GTPase activity of NG-FlhF, in stark contrast to the FlhF-GTPase stimulating protein FlhG (Fig. 3e). Taken together, we show that the NG domain of FlhF interacts with the C-terminal cytoplasmic region of HubP in an apparently nucleotide-independent manner without affecting the GTPase activity of FlhF.

## FlhF can bring FliG into the proximity of HubP
We have shown that FlhF can interact with the C-ring protein FliG and the polar landmark protein HubP through its FID- and NG-domains, respectively. In a next step, we wanted to study whether both interactions would be possible at the same time. Therefore, StrepII-tagged HubP-C was used as bait and FlhF and FliG as prey. As shown above, FlhF interacted with HubP-C, and when FliG was added a stoichiometric complex of the three proteins was observed (Fig. 3f). This result shows that FlhF is able to bridge HubP and FliG in vitro. These data allow a hypothesis in which the NG domain of FlhF mediates interaction with the polar landmark HubP, while the first 44 N-terminal residues of FlhF B-domain interact with FliG to initiate flagellar formation.

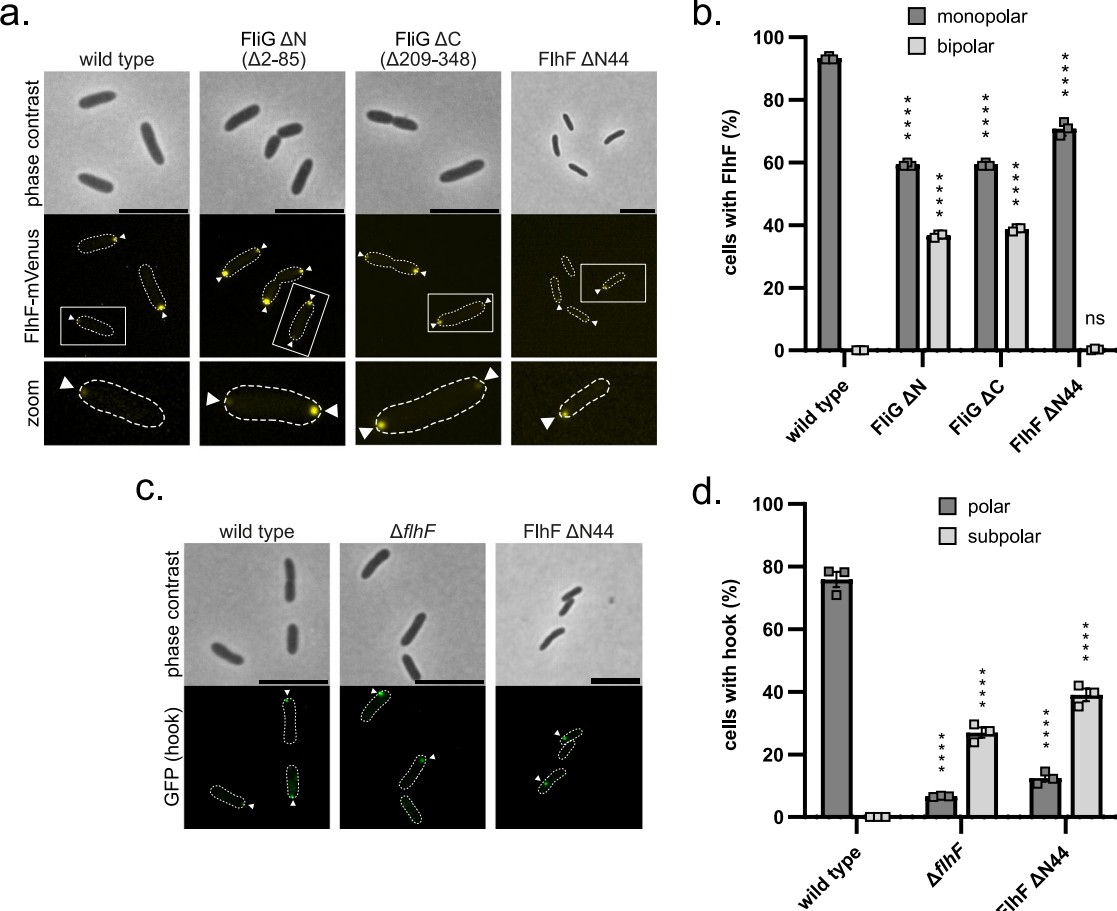

**Fig. 4 | The 44 aa N-terminal domain is required for FlhF function in Shewanella. a** Microscopic images of the indicated *S. putrefaciens* strains expressing FlhF-mVenus. The top row shows the phase contrast images; the row below shows the corresponding fluorescence images. The panel at the bottom displays an enlargement of the boxed areas within the fluorescence images. Fluorescent FlhF-mVenus foci are marked with a white arrow. The scale bar equals 3 μm. **b** Quantification of the FlhF-mVenus localization patterns in *S. putrefaciens* based on the microscopy images shown in **a**. Data was obtained from three individual experiments (*n* = 3) where at least 327 cells were counted for each biological replicate of each strain (****$p$ < 0.0001, two-way ANOVA, ns = not significant; here $p$ = 0.9663). Data are presented as mean values together with the corresponding ± standard error of mean (SEM). **c** Microscopic images of hook stains from the indicated *S. putrefaciens* strains with Alexa Fluor 488-C5-maleimide dye. The top row shows the phase contrast images, while the lower row shows the corresponding fluorescence images. Fluorescent hooks are marked with a white arrow. The scale bar equals 3 μm. **d** Quantification of the hook localization pattern in *S. putrefaciens* based on the microscopy images shown in Fig 4c. Data was obtained from three individual experiments (*n* = 3) where at least 330 cells were counted for each biological replicate of each strain (****$p$ < 0.0001, two-way ANOVA). Data are presented as mean values together with the corresponding ± standard error of mean (SEM). Source data are provided in the Source Data file.

If correct, we anticipate that a FlhF variant lacking its N-terminal FliG-binding region would still localize to the cell pole, while simultaneously loosing capability to recruit FliG. We therefore determined the localization of a ΔN44-FlhF mutant in vivo and its effect on flagellar positioning. For this purpose, we utilized a *S. putrefaciens* strain with a chromosomal fusion of mVenus to *flhF* (*flhF*-mVenus)[16], wherein we specifically deleted the N-terminal 44 residues of the *flhF* gene (*flhF*ΔN44-mVenus). Furthermore, we specifically labeled the hook structures of the primary polar flagellar system of the strain by introducing a T183C substitution in the flagellar hook protein FlgE1, allowing for the coupling of maleimide-ligated fluorescent dye[18,28,29]. Importantly, the N-terminally truncated FlhF-mVenus was consistently produced at levels comparable to the wild-type (Supplementary Fig. 2).

Fluorescence microscopy revealed fewer cells producing ΔN44-FlhF-mVenus displayed fluorescent foci (about 70% compared to about 90% of full-length FlhF-mVenus; Fig. 4a). However, these foci were always located at the cell pole (Fig. 4a, b). In contrast, in the majority of ΔN44-FlhF mutant cells single flagellar hooks appeared in subpolar/lateral positions (about 40%, 10% polar; Fig. 1c,

Fig. 4c, d), while they exclusively appeared at the cell pole in about 75% of wild-type cells (Fig. 4c, d). Accordingly, a ΔN44-FlhF mutant phenocopies a *ΔflhF* mutant with respect to spreading through soft agar (Supplementary Fig. 3). The analysis confirmed the hypothesis that ΔN44-mutants of FlhF retain their ability to localize FlhF to the cell pole, but uncouple FlhF localization from that of the flagella machinery.

Vice versa, we also tested whether localization of FlhF is affected in the presence or absence of FliG as interaction partner. To this end, we used fluorescence microscopy on a *S. putrefaciens* strain producing mVenus-labeled FlhF (FlhF-mVenus) bearing an N-terminal (Δ2-85 aa; FliG ΔN) or a C-terminal (Δ209-348 aa; FliG ΔC) deletion in FliG. Both deletions in FliG resulted in a pronounced accumulation of FlhF-mVenus at one (about 60%) or both cell poles (Fig. 4a, b) compared to wild type-background, which exclusively exhibited monopolar localization in about 92% of the cells (Fig. 4b). The amount of polar fluorescence is also reflected in the amount of FlhF-mVenus protein produced in the cells (Supplementary Figs. 4a, b). These findings suggest that coupling to FliG and/or initiation of flagellar assembly affects FlhF accumulation at the cell pole.

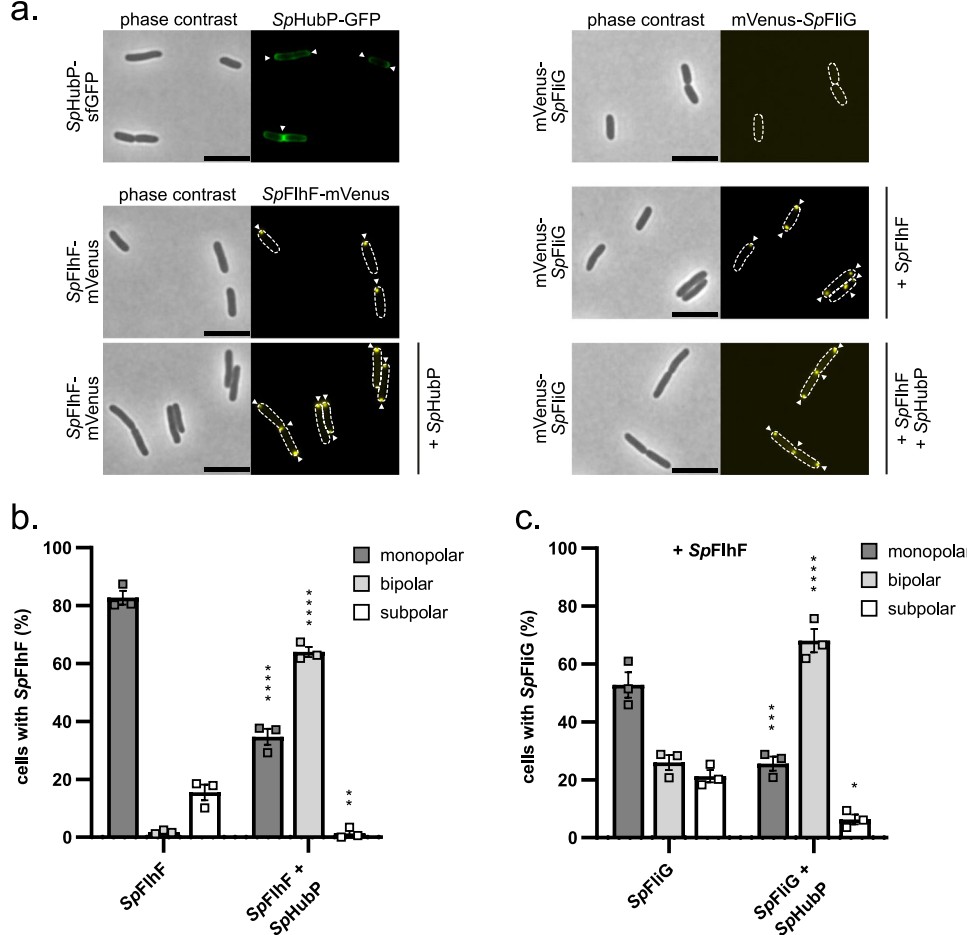

**Fig. 5 | The Shewanella HubP-FlhF-FliG recruitment cascade can be rebuilt in *Escherichia coli*. a** Microscopic images of the indicated E. coli DH5α strains containing the expression plasmids for the wild-type version of HubP (SpHubP) or fluorescently tagged versions of *S. putrefaciens* HubP (SpHubP), FlhF (SpFlhF) or FliG (SpFliG). The left row shows the phase contrast images, while the right row shows the corresponding fluorescence images. Fluorescent foci of the respective fluorescently labeled protein are marked with a white arrow. The scale bar equals 3 µm. **b**, **c** Quantification of the localization pattern of *Sp*FlhF (**b**) or *Sp*FliG (**c**) with *Sp*FlhF in *E. coli* DH5α based on the previous microscopy images. Data was obtained from three individual experiments ($n = 3$) where at least 310 cells were counted for each biological replicate of each strain (****$p < 0.0001$, ***$p = 0.0001$, **$p = 0.0011$, *$p = 0.014$; two-way ANOVA). Data are presented as mean values together with the corresponding ± standard error of mean (SEM). Source data are provided in the Source Data file.

## The Shewanella HubP-FlhF-FliG recruitment cascade can be rebuilt in *E. coli*

So far, our findings suggested that, in *S. putrefaciens*, flagellar synthesis is initiated with FliF being localized to the cell pole by the polar landmark protein HubP, to where it then recruits FliG. If correct, this recruitment cascade may also be rebuilt in *Escherichia coli*, which lacks orthologs to HubP and FlhF. To test this, fluorescently tagged or wild-type versions of HubP, FlhF and FliG were ectopically produced either alone or in combinations from suitable expression plasmids in *E. coli* DH5α. As previously observed[16], HubP-sfGFP predominantly localized to the polar regions of the cells and particularly accumulated in the cell division plane (Fig. 5a). FlhF-mVenus produced alone formed small monopolar and sometimes subpolar clusters in the cells (compare also to Fig. 1c). However, when expressed in concert with HubP, FlhF also appeared bipolarly and frequently occurred in the cell division planes, which was never observed in the absence of HubP (Fig. 5a, b). mVenus-FliG expressed alone did not give any detectable fluorescence signal. In the presence of FlhF, small clusters of mVenus occurred in mono-, bi- and subpolar positions. In the additional presence of HubP, mVenus-FliG formed clusters at the cell poles and division planes (Fig. 5a, c). As the proteins are not produced at their physiological levels in a heterologous host, we do not presume that FliG self-assembles into larger structures. Nevertheless, the observed localization pattern strongly suggests that HubP is capable of recruiting both proteins in *E. coli* as well. The absence of a mVenus-FliG fluorescent signal in the absence of FlhF suggested that the latter may stabilize FliG. Correspondingly, western blotting showed that mVenus-FliG is only stable in the presence of FlhF, but not alone or in the presence of HubP (Supplementary Fig. 5), at least in the heterologous host *E. coli*. This may suggest that FliG requires stabilization by FlhF during flagellar assembly.

## Discussion

In this study, we aimed at gaining a deeper mechanistic understanding of how the SRP-GTPase FlhF enables the polar localization of a flagellum in the polar flagellated bacteria, such as *S. putrefaciens* or Vibrio species. Previous studies have indicated that FlhF might act in the context of the flagellar C-ring, and proposed that FlhF establishes the site of flagellum assembly at the old cell pole membrane by recruiting the earliest flagellar structural component FliF[11,12,30]. It might therefore be involved in assembly of the flagellar C-ring[31]. However, the precise mechanism by which FlhF acts in the context of C-ring assembly, and whether FlhF would directly or indirectly interact with FliF was not known. Furthermore, it raised the question of whether any interaction between FlhF and the flagellar C-ring alone is adequate to fully explain how FlhF establishes polar localization of the flagellum.

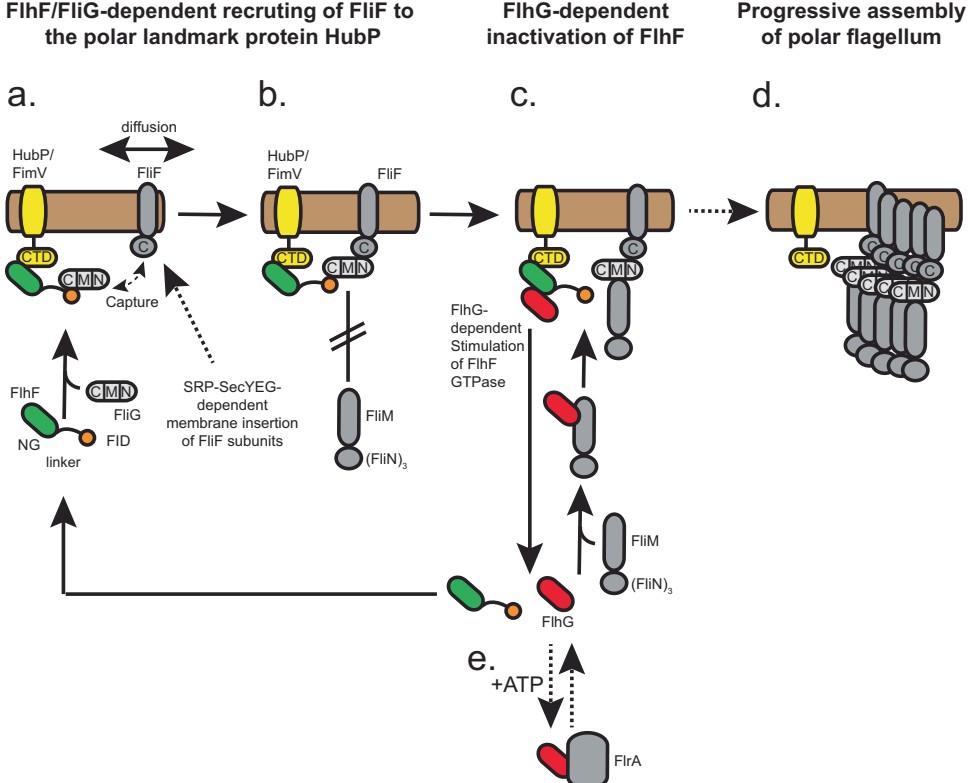

**Fig. 6 | Model describing how FlhF establishes flagellar localization of polar flagella. a** The FID domain of FlhF binds to FliG, while the NG domain of FlhF interacts with the CTD of HubP. This configuration allows the FiD-bound FliG to capture the transmembrane protein FliF through its C-terminal domain. **b** FlhF hinders the interaction between FliG and its C-ring partners FliM/FliN, while permitting FliG to engage with FliF. **c, d** FlhG stimulates the GTPase activity of FlhF, initiating another round of FlhF-mediated diffusion-capture, which leads to the successive assembly of the C-ring. **e** When not engaged with FliM/FliN, FlhG can interact with the master transcriptional regulator FlrA. The color code is: FlhF with its NG, linker and FID domains (green, black and orange, respectively), HubP/FimV (yellow), flagella building blocks (grey tones), FlhG (red) and the cytoplasmic membrane (light brown). Further descriptions are given in the discussion.

In this study, we demonstrate that the multidomain protein FlhF, comprising an N-terminal B-domain followed by an SRP-type GTPase domain (aka: NG domain), both being connected by a linker region (Fig. 2b), can serve as a tether between the polar landmark protein HubP/FimV and the developing flagellar structure. While the NG domain of FlhF interacts with the C-terminal domain of the landmark protein HubP, a structured domain at the very N-terminus of its B-domain interacts with the flagellar C-ring protein FliG (Fig. 6a). This domain at the N-terminus of FlhF, which establishes the FlhF-FliG interaction, was termed the FliG interaction domain (FID). In vitro, FID mainly interacts with the middle and C-terminal (MC) domain of FliG. Notably, a truncation of the N-terminal or C-terminal region both resulted in a bipolar localization of FlhF in vivo (Fig. 4a, b). The reason for this remains elusive so far. As HubP occurs at both cell poles in *S. putrefaceins*[16], a highly interesting hypothesis is that FlhF bound to non-functional FliG loses the specificity for designated cell pole. However, this remains speculative and requires further studies.

We propose a model where FID-tethered FliG "catches" membrane-diffusing FliF proteins, which are likely being inserted into the membrane in an SRP-dependent co-translational manner via the canonical SecYEG pathway (reviewed in[32,33]:). Whether the observation that FlhF constitutes the third member of SRP-GTPase family besides Ffh and FtsY is of functional relevance remains to be seen. So far, we have no reasonable evidence to believe that FlhF directly interacts with the SRP system, or serves in the co-translational insertion of any membrane protein. However, and given the uniform distribution of SecYEG machines along the cytoplasmic membrane[34], insertion of FliF into the membrane can occur in close proximity to the pole (Fig. 6a). Hence, the suggested "diffusion-capture" mechanism, wherein FlhF

anchors FliF via FliG and HubP to the cell pole, may primarily function to retain nascent flagellar building blocks at the cell pole and prevent their dispersion away from this crucial assembly site (Fig. 6a, b). This view is supported by our finding that removal of the FID instantaneously leads to delocalized flagella.

A surprising finding was that while the FlhF-FliG interaction allows engagement of FliG to FliF, it does not permit the interaction of FliG with its C-ring partners FliM/FliN (Fig. 6b). Thus, the interaction of FlhF and FliG provides an impediment for the latter to complete C-ring assembly via FliM and FliN. This feature could provide a checkpoint: FlhG, which stimulates the GTPase activity of FlhF[15–17,35], interacts with FliM/FliN via the N-terminus of the FliM protein[19,20]. Thus, we envision that FliM/N complexes are only admitted to FlhF-bound FliF-FliG complexes when FlhG is present (Fig. 6c). In such a way, FlhG can stimulate the GTPase activity of FlhF to initiate a further round of FlhF-mediated "diffusion-capture" of the next flagellar building blocks to the cell pole via the landmark protein HubP (Fig. 6).

Our previous findings in which FlhG connects flagellar C-ring assembly with the activity of the master transcriptional regulator FlrA[20] align with the proposed model (Fig. 6e). FlrA and FliM share a binding site on FlhG, with their interaction dependent on FlhG's ATP-dependent dimerization state; FliM binds independently of nucleotide binding, whereas FlrA exclusively interacts with the ATP-bound FlhG dimer, stimulating its ATPase activity[20]. FlhG's recruitment to the cell pole is contingent upon the advancement of C-ring assembly, thereby affecting its interactions with both the master transcriptional regulator FlrA and the GTPase FlhF. In conclusion, our study elucidates the molecular framework governing how FlhF coordinates the polar localization of the flagellum, working closely with the polar landmark

protein HubP and facilitating the assembly of flagellar MS-ring/C-ring components at the cell pole.

How conserved is the HubP-FlhF-dependent pathway for polar flagella localization? Interestingly, in species with a corresponding ortholog, the role of HubP in flagellation is not consistent even among bacteria of the same genus. In *Vibrio cholerae*, the loss of HubP has only a minor effect on flagellation[26], whereas in *V. parahaemolyticus*, HubP deletion results in the loss of flagellation in about half of the cells[25]. In contrast, in *V. alginolyticus*, the number of polar flagella increases when HubP is deleted[24]. Notably, in *S. putrefaciens*, the loss of HubP has little effect on flagella polarity[16]. These differences suggest that HubP functions within a more complex network during polar flagella assembly, and other factors may also play a contributory role.

Indeed, a significant determinant for polar flagellation, the transmembrane FlhF-interacting protein FipA, was recently identified[18]. FipA is present in all bacterial species that rely on the FlhF/FlhG system to organize polar flagellation. In *V. alginolyticus*, *Pseudomonas putida*, and *S. putrefaciens*, HubP (or FimV in *P. putida*) and FipA work together to recruit FlhF to the cell pole. However, the specific roles of these proteins in the biogenesis of polar flagella seem to have diversified among the three species. In *S. putrefaciens*, HubP and FipA appear to be nearly redundant in function[18]. The interactions and regulatory effects among HubP, FipA, and FlhF are currently being investigated.

Other polarly flagellated bacteria can lack homologs to HubP and FlhF entirely. The best-studied example among these species is the alphaproteobacterium *Caulobacter crescentus*. In *C. crescentus*, the flagellum's position is defined by the polar landmark protein TipN[36,37]. In a cell cycle-dependent manner, TipN recruits another transmembrane protein, TipF, to the flagellar assembly site. TipF further recruits a third flagellar polarity factor, PflI, and also directly interacts with the flagellar protein FliG[8,38,39]. Thus, *C. crescentus* uses a three-step cascade to initiate polar flagellar assembly, similar to that described for *S. putrefaciens*, but relies on different localization factors. This suggests that while the mechanisms of recruiting polar flagella vary among species, the general principle of using specific localization factors is widely conserved.

## Methods

### Protein production and purification

Gene fragments encompassing the employed proteins and their variants were amplified by polymerase chain reaction and inserted into a pET24d vector (Novagen) via NcoI/XhoI restriction sites. The accession codes of the genes encoding for the employed proteins are: *flhF*: Sputcn32_2561, *fliG*-polar: Sputcn32_2575, *fliG*-lateral: Sputcn32_3475, *flhG*: Sputcn32_2560, *fliM*: Sputcn32_2569, *fliN*: Sputcn32_2568, *fliF*: Sputcn32_2576 and *hubP*: Sputcn32_2442.

*Escherichia coli* strain BL21 (DE3) (Novagen) was employed for protein production, with cells cultured in lysogeny broth medium (LB) supplemented with 1.5% (w/v) d(+)-lactose monohydrate for 16 h at 303 K. Cell pellets were resuspended at a ratio of 10 ml of lysis buffer per gram of cells and then subjected to processing through an M1-10L Microfluidizer (Microfluidics). The lysis buffer, comprising 20 mM Na-HEPES at pH 8.0, 250 mM NaCl, 10 mM $MgCl_2$, and 10 mM KCl, was employed for this purpose. The resulting lysate underwent clarification through centrifugation (125,000 g for 30 min at 277 K) using a Ti-45 rotor (Beckmann) and was subsequently applied to a 1 ml HisTrap HP column (GE Healthcare). The column underwent an initial wash with five column volumes of lysis buffer containing 40 mM imidazole at pH 8.0. Protein elution was carried out in lysis buffer containing 500 mM imidazole at pH 8.0. Following elution, the protein was concentrated to approximately 30 mg/ml using an Amicon Ultracel-10K (Millipore) and then subjected to size-exclusion chromatography using either an S75/26–60 column or an S200/26–60 (GE Healthcare) in the same buffer as before but

without imidazole. Fractions containing the protein were combined and concentrated as required.

### Pulldown assays

StrepTagII pulldown assays conducted in order to study protein interactions. Therefore, StrepII-tagged protein cultures and tested His-tagged protein cultures (400 ml) were combined and lysed following the procedures outlined above. Subsequently, the lysates were incubated with 30 µl of MagStrep Strep-Tactin XT beads (iba Life Sciences) for 30 min at 4 °C with gentle rotation. Following centrifugation (4000 rpm, 5 min, 4 °C), the supernatant was discarded, and the beads underwent three washes with 500 µl of SEC Buffer, utilizing a magnetic rack. The proteins bound to the beads were eluted using 200 µM D-Biotin in SEC buffer and then subjected to analysis through SDS-PAGE.

### GTPase activity assay

To assess the impact of interaction partners on the GTPase activity of FlhF, only proteins (including NG-FlhF) purified through size-exclusion chromatography were employed. Specifically, 1 nmol of NG-FlhF was incubated either alone or with 2 nmol of FlhG or HubP-C in a total reaction volume of 50 µL. The GTP concentration was 2 mM. The reaction proceeded for 60 min at 37 °C without shaking. The reaction was stopped by the addition of 100 µl chloroform to each reaction, followed by boiling for 15 s at 98 °C and rapid freezing in liquid nitrogen. Thereafter, each sample was thought and cleared by centrifugation (i.e., 13,000 RPM for 15 min in a benchtop centrifuge). Subsequently, the samples were analyzed by high-performance liquid chromatography (HPLC) on an Agilent 1260 Series system (Agilent Technologies) equipped with a Metrosep A Supp5 – 150/4.0 column (Metrohm International). The HPLC buffer, with a pH of 9.25 and comprising 90 mM $(NH_4)_2CO_3$, flowed at a rate of 0.6 ml/min. Nucleotides were detected at 260 nm.

### Crystallization and structure determination

Crystallization was executed using the sitting-drop method at 20 °C with 250-nL drops containing an equal mixture of 1 mM protein and precipitation solutions. The specific crystallization conditions were 1.6 M sodium citrate, pH 6.5. Data collection took place under cryogenic conditions at the P13 beamline, Deutsches Elektronen Synchrotron (DESY, Hamburg, Germany). Subsequently, the collected data were processed using XDS and scaled with XSCALE[40]. Structural determination involved molecular replacement with PHASER[41], utilizing the relevant part of the *Sp*FlhF Alphafold model[42] with the accession code AF-A4Y8J9-F1. Manual building was carried out in COOT[43], and refinement was conducted using PHENIX 1.18.2[44].

### Yeast two-hybrid analysis

For Y2H interaction assays, plasmids expressing the FlhG bait protein, fused to the Gal4 DNA-binding domain, and prey proteins, fused to the Gal4 activation domain, were co-transformed into the reporter strain PJ69-4A (*MAT**a** trp1-901 leu2-3,112 ura3-52 his3-200 gal4Δ gal80Δ LYS2::GAL1-HIS3 GAL2-ADE2 met2::GAL7-lacZ*)[45]. Y2H interactions were documented by spotting representative transformants in 10-fold serial dilution steps onto SC-Leu-Trp (-LT), SC-His-Leu-Trp (-HLT; *HIS3* reporter), and SC-Ade-Leu-Trp (-ALT; *ADE2* reporter) plates, which were incubated for 3 day at 30 °C. Growth on -HLT plates is indicative of a weak or moderate interaction, and only relatively strong interactions also permit growth on -ALT plates.

### Western blot analysis

Western blot analysis was performed to check the stability and expression of the fusion proteins. The protein lysates of the respective strains were obtained from an exponentially growing culture and adjusted to the same optical density (OD600 of 10). For separation by

SDS-PAGE, 10 µl of the samples were loaded onto the SDS-gel. The protein extracts were then transferred to membranes and visualized by Western blotting with antibodies against GFP or FlhF[22]. The respective antibodies are coupled to AP and CDP-Star chemiluminescent substrate (Roche, Switzerland) was used to generate a luminescent signal. The signal was detected using a Fusion-SL chemiluminescence imager (Peqlab, Erlangen, Germany). Uncropped and unprocessed scans of the most important blots can be found in the Source Data file.

## Growth conditions and media

For all cloning experiments, *E. coli* cells were grown in LB medium or LB agar plates at 37 °C containing antibiotics of the following concentrations: 50 µg/ml kanamycin, 30 µg/ml chloramphenicol. Ectopic expression was induced during exponential growth for 1 h from pBAD or pBBR-derived plasmids with 0.05% L-arabinose and 0.5 mM IPTG. *S. putrefaciens* cells were grown in LB medium or LB agar plates at 30 °C. If necessary, media supplemented with 50 µg/ml kanamycin, 300 µM 2,6-diaminopimelic acid, and/or 12% (w/v) sucrose were used for conjugation.

## Strain constructions

The bacterial strains and plasmids used in this study are listed in Supplementary Tables 2, 3 and 4, respectively. The primers used are indicated in Supplementary Table 5. To introduce DNA into *S. putrefaciens*, *E. coli* WM3064 was used. *E. coli* DH5αλpir was used for cloning and experiments. For chromosomal deletions in *S. putrefaciens* sequential crossover was conducted as previously described[16] using derivatives of the plasmid pNPTS138-R6K[46]. Corresponding plasmids were constructed by Gibson assembly[47] by combining PCR-derived fragments with EcoRV-digested pNPTS138-R6K.

## Hook staining

For staining of extracellular hook structures of *S. putrefaciens* a T183C substitution was introduced into the gene sequence encoding the FlgE hook protein. The strains were harvested from an exponentially growing culture and always handled with cut pipette tips to avoid shear forces on the extracellular structures. After gentle centrifugation at 1200 g for 5 min, the cell pellet was resuspended in 50 µl of 1× PBS. For staining, a maleimide ligate dye (Alexa Fluor 488-C5-maleimide fluorescent dye; Thermo Fisher Scientific) was added and incubated in the dark for about 20 min. Afterwards cells were carefully washed twice with 1x PBS to remove unbound ligate dye. The desired structures were detected as described in the fluorescence microscopy section.

## Microscopy

For imaging of samples, 2 µl of the respective strain were spotted on a 1% PBS-agarose (select agar, Invitrogen). Fluorescence microscopy was performed as described previously[48], using a microscope set-up based on a Leica DMI 6000 B inverse microscope (Leica), equipped with a pco.edge sCMOS camera (PCO), a SPECTRA light engine (lumencor), an HCPL APO 63×/1.4−0.6 objective (Leica) using a custom filter set (T495lpxr, ET525/50 m; Chroma Technology) and the VisiView software (Visitron Systems, Puchheim, Germany). Microscopy images were analyzed by using ImageJ (v1.54 g). Statistics and graph creation were done using Prism 9.5.1 (GraphPad software). Foci intensity analysis was made using BacStalk 1.8stable[23].

## Soft-agar spreading assays

For *S. putrefaciens* soft-agar spreading assays, 2 µl of an exponentially growing culture were spotted onto 0.25% LB agar plates (select agar, Invitrogen). Plates were incubated for about 18 h at 30 °C. For documentation, plates were scanned using an Epson V700 photo scanner. Different strains were always spotted on the same plate to ensure a direct comparison.

## Reporting summary

Further information on research design is available in the Nature Portfolio Reporting Summary linked to this article.

## Data availability

Coordinates and structure factors of the crystal structure of FID have been deposited at the Protein Data Bank with the accession code: 9EN1. Source data are provided with this paper. The PDB code of the FlhF-GTPase bound to GDP is 8R9R. Source data are provided with this paper.

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

## Acknowledgements

We thank the Deutsche Forschungsgemeinschaft (DFG) for financial support (Grant number: 495924434 to G.B. and K.M.T). We would like to thank Devid Mrusek and Guillaume Murat for their contributions in the beginning of the project. We acknowledge the technical support by Ulrike Ruppert. G.B. would like to thank I. Sinning (Heidelberg) for ongoing support.

## Author contributions

AD, CNM, and JPF conducted the in vitro experiments. AD, CNM, and GB carried out the structure analysis. LMS, JCH, and KT performed the in vivo studies. DK and GB conducted the Y2H analysis. GB and KT designed and supervised the research. AD, LMS, KT, and GB wrote the manuscript with input from all authors. GB and KT secured funding for the study.

## Funding

## Competing interests

The authors declare no conflicts of interest.
