## [Peer Review File · Nature Communications]

Polar confinement of a macromolecular machine by an SRP-type GTPaseREVIEWER COMMENTS

Reviewer #1 (Remarks to the Author):

In this manuscript, using polarly flagellated bacteria, *Shewanella putrefaciens* CN32, authors for the first time to demonstrate that FlhF, the flagellar biosynthesis factor, binds to flagellar C-ring protein FliG only but not FliM/FliN, at its relatively uncharacterized N-terminal region. They also found that the NG-domain of FlhF binds to the cytoplasmic region of HubP in nucleotide (GTP/GDP)-independent manner. The X-ray structural analysis of FlhF B-domain and in vitro binding experiments were well-combined with in vivo protein localization analyses in native *Shewanella* background, revealing that C-ring protein FliG is recruited (or localized) to the cell pole via HubP-tethered FlhF. Furthermore, authors demonstrated the reconstitution of the FlhF/HubP-dependent polar localization of *Shewanella* FliG in *E. coli* cells. Based on these results, they proposed a model of polar confinement of the flagellar biogenesis. This reviewer enjoyed reading this beautiful manuscript: another successful collaboration of Bange lab and Thormann lab. Data shown here are clearly presented, and texts are well organized and described. I have some concerns, relatively minor, which are listed below:

Major points:

- 1) It is unfortunate that authors do not discuss about generality of the proposed model in Fig. 6 among other polar flagellated bacterial species, presumably because of the word counts for the main text. But still, I would like to know the authors opinion whether the model can be limited to the *Shewanella putrefaciens* CN32, or more generally applicable to other polarly flagellated species. Such a discussion can be included in "supplementary discussion". In fact, effect of HubP on the polar flagellation/motility is diverse among species. For example, lack of HubP only moderately affects flagellation in *Vibrio cholerae*, whereas in *Vibrio alginolyticus* it causes multiple flagellation.
- 2) In Fig. 4C, $\Delta flhF$ caused hook formation in the subpolar location. I assume FlgE1 is encoded in the flagellar cluster 1 and included only in the Na⁺-driven polar flagellum, but would it be possible to be included in the lateral flagellar system? When authors observed $\Delta flhF$ strain under the microscope (light or electron microscopy), are the flagella formed at the lateral position? I wonder, the subpolar localization of the hook protein FlgE1 shown in Fig. 4C reflects, whether the formation of primary flagellum (cluster 1) at the lateral position without FlhF, or the secondary flagellar formation that happened to include FlgE1 at the hook. When the FlgE1-T183C expressed in the background of Δ cluster 2, did the author only observe the polar cluster of fluorescence?
- 3) I would like to know authors' idea about how the multimeric? structure of polar localized FliG look like, tethered by HubP and FlhF at *E. coli* cell. Fluorescent foci must reflect multiple FliG molecules at the cell pole, and in this case, there are no *Shewanella* FliF protein expressed, meaning no MS ring platform at the pole. I do not think FliG forms the ring.
- 4) In the model (Fig. 6), panel D did not contain FlhF (and FlhG). Does this mean FlhF dissociates from the HubP after FlhF GTPase activity is stimulated by FlhG? I raised this question because FlhF still can be observed at the flagellated cell pole after the polar flagellation is completed, at least in *Vibrio* species.
- 5) Would you explain to me how does FlhG control flagellar number negatively? Does the negative regulation by FlhG not occur in *Shewanella*? In the proposed model, FlhG is necessary to assemble the C-ring via recruiting FliM/N to the HubP/FlhF-anchored FliF/G complex. Since deletion of FlhG increases the number of flagella at cell pole in *Vibrio* species, I am somewhat confused. This question is beyond the scope of this manuscript, but I am just curious to know the authors opinion.

Minor point

- 6) Introduction, line 22. It would be good to have one more reference that describes stimulation of FlhF GTPase by FlhG: Gulbranson et al, Mol. Microbiol. 2016.
- 7) Fig. 1b, legend: I assume this is FlhF of *Shewanella putrefaciens*, and source of protein should

be noted.

8) Fig. 1c and in the text. For the general readers of Nature Communications, it would be good to clearly stated how different between "sub"-polar location and "lateral"-location.

9) Fig. 2b, lower left panel. In page 3, line 27; I wonder, which constructs did authors used for the X-ray crystal structure analysis. Did authors use pET24d_1-60N-Strep or pET24d_FlhF_1-137_N-Strep? I raised this question is because, I wonder whether following residues C-terminal to the residue 46 is non-structured (disordered) in the crystal or the construct does not contain those residues.

10) Fig. 4a, lower panels. For me, it is a little bit difficult to see the bipolar localization of FlhF-mVenus. For example, the fluorescent dot at the bottom of the rightmost cell in FliG Δ N/FlhF-mVenus image is so faint, even though it is highlighted by the arrowhead.

11) page 5, line 47; Did author not detect any fluorescence signal from the cells expressing mVenus-FliG alone? If so, it would be good to state that facts.

12) In Fig. 5a, would it be possible to distinguish between the subpolar foci and bipolar foci by the color or symbol? It may be good for the general readers.

13) Discussion, line 13, "Vibrio" should be italic.

14) Discussion, line 38, I do not catch why authors assume "reasonable" for the polar insertion of FliF. In the previous phrase, it was stated that SecYEG machine is uniformly distributed.

15) Fig. 1, legend, line 22; proteinS

16) no page numbers were provided for reference 7, 10 and 27.

Reviewer #2 (Remarks to the Author):

FlhF and FlhG control the location and number of flagella in many polar flagellated bacteria, although the mechanisms by which these proteins affect the flagellation pattern are poorly understood. The authors show here that the GTPase (NG) domain of FlhF interacts with the polar landmark protein HubP and a region within the N-terminal B-domain of FlhF (residues 1-46; FliG interaction domain, FID) binds FliG in *Shewanella putrefaciens*. FlhF binds specifically to the FliG associated with polar flagella as it does not bind to the FliG paralog that forms lateral flagella. The authors also show that FlhF binds FliG and HubP simultaneously. They propose a model where the FlhF-FliG complex binds HubP at the cell pole and FliF is then recruited to the pole to facilitate formation of the MS-ring. The data presented in the manuscript provide new information on addressing the mechanism by which FlhF assists in targeting flagella to the cell pole, at least for bacterial species that have a HubP homolog.

Specific comments:

1. HubP/FimV homologs are relatively limited in their phylogenetic distribution, which raises the question how applicable the proposed model is for polar flagellated bacteria that lack HubP homologs. It would enhance the impact of the manuscript if the authors could demonstrate that a FlhF protein from bacterial species that lacks a HubP homolog also binds FliG.

2. It would be informative if the authors included data on the localization of HubP in *S. putrefaciens*. Does HubP localize to both cell poles in *S. putrefaciens* as the authors show it does in *Escherichia coli* in Figure 5a? If HubP localizes to both cell poles in *S. putrefaciens*, some discussion of what might prevent the FlhF-FliG complex from binding to HubP at both cell poles would be helpful.

3. It is not clear how the observation that the FlhF-mVenus protein localizes to both cell poles when the N-terminus or C-terminus of FliG is deleted fits into the model shown in Figure 6 since the FlhF Δ N44 variant localizes exclusively to one cell pole. Given that the FlhF variant does not bind FliG, it is not obvious how the truncated FliG proteins affect the localization of the FlhF Δ N44 variant.

Minor comments:

1. Update reference 17 and change the name FIP to FipA to reflect the updated reference.

2. Additional details in legend to Figure 1 would be helpful. For example, explain the HIS3 and ADE2 reporters. Also, why does the colony color vary among the strains and growth media?

REVIEWER COMMENTS

Reviewer #1 (Remarks to the Author):

In this manuscript, using polarly flagellated bacteria, *Shewanella putrefaciens* CN32, authors for the first time to demonstrate that FlhF, the flagellar biosynthesis factor, binds to flagellar C-ring protein FliG only but not FliM/FliN, at its relatively uncharacterized N-terminal region. They also found that the NG-domain of FlhF binds to the cytoplasmic region of HubP in nucleotide (GTP/GDP)-independent manner. The X-ray structural analysis of FlhF B-domain and in vitro binding experiments were well-combined with in vivo protein localization analyses in native *Shewanella* background, revealing that C-ring protein FliG is recruited (or localized) to the cell pole via HubP-tethered FlhF. Furthermore, authors demonstrated the reconstitution of the FlhF/HubP-dependent polar localization of *Shewanella* FliG in *E. coli* cells. Based on these results, they proposed a model of polar confinement of the flagellar biogenesis. This reviewer enjoyed reading this beautiful manuscript: another successful collaboration of Bange lab and Thormann lab. Data shown here are clearly presented, and texts are well organized and described. I have some concerns, relatively minor, which are listed below:

We thank this reviewer for the very positive evaluation of our manuscript and the relevant points raised.

Major points:

1) It is unfortunate that authors do not discuss about generality of the proposed model in Fig. 6 among other polar flagellated bacterial species, presumably because of the word counts for the main text. But still, I would like to know the authors opinion whether the model can be limited to the *Shewanella putrefaciens* CN32, or more generally applicable to other polarly flagellated species. Such a discussion can be included in "supplementary discussion". In fact, effect of HubP on the polar flagellation/motility is diverse among species. For example, lack of HubP only moderately affects flagellation in *Vibrio cholerae*, whereas in *Vibrio alginolyticus* it causes multiple flagellation.

*This reviewer raises an important point. The apparently differing roles of HubP in various species are a major reason we were cautious about proposing overly general mechanisms. In a recent collaboration, the Thormann group identified another polar determinant for FlhF, the transmembrane protein FipA. Studies on three different model species demonstrated that HubP/FimV and FipA interact in recruiting FlhF. Notably, in *S. putrefaciens*, both HubP and FipA may have interchangeable functions in this process. We have updated the Discussion section to reflect these findings (second to last paragraph).*

2) In Fig. 4C, Δ flhF caused hook formation in the subpolar location. I assume FlgE1 is encoded in the flagellar cluster 1 and included only in the Na⁺-driven polar flagellum, but would it be possible to be included in the lateral flagellar system? When authors observed Δ flhF strain under the microscope (light or electron microscopy), are the flagella formed at the lateral position? I wonder, the subpolar localization of the hook protein FlgE1 shown in Fig. 4C reflects, whether the formation of primary flagellum (cluster 1) at the lateral position without FlhF, or the secondary flagellar formation that happened to include FlgE1 at the hook. When the FlgE1-T183C expressed in the background of Δ cluster 2, did the author only observe the polar cluster of fluorescence?

*Thanks for pointing out that possibility. We have already used the FlgE substitutions in other studies on *S. putrefaciens*. FlgE1 of the polar system never occurs at lateral positions, and conversely, FlgE2 of the secondary system only occurs at lateral positions (see Rossmann et al., 2019, J Bacteriol 201; doi: 10.1128/JB.00534-18). In the absence of the lateral system, FlgE1 is exclusively found at the cell pole (see Hook et al., 2020, Front Microbiol 11: 564161; doi: 10.3389/fmicb.2020.564161). Additionally, the role of FlhF in placing the polar flagellum in *S. putrefaciens* is described in detail in Rossmann et al., 2015, Mol Microbiol 98: 727; doi: 10.1111/mmi.13152. We have added these references where appropriate.*

3) I would like to know authors' idea about how the multimeric? structure of polar localized FliG look like, tethered

by HubP and FlhF at E. coli cell. Fluorescent foci must reflect multiple FliG molecules at the cell pole, and in this case, there are no Shewanella FliF protein expressed, meaning no MS ring platform at the pole. I do not think FliG forms the ring.

The reviewer is correct; we do not assume that FliG forms any ring-like structures. Our primary goal in this experiment was to demonstrate that the recruitment cascade can be reconstructed in a species lacking HubP or FlhF, which we successfully achieved. We have added the following to the last paragraph of the Results section: "...As the proteins are not produced at their physiological levels in a heterologous host, we do not presume that FliG self-assembles into larger structures. Nevertheless, the observed localization pattern strongly suggests that HubP is capable of recruiting both proteins in E. coli as well. ..."

4) In the model (Fig. 6), panel D did not contain FlhF (and FlhG). Does this mean FlhF dissociates from the HubP after FlhF GTPase activity is stimulated by FlhG? I raised this question because FlhF still can be observed at the flagellated cell pole after the polar flagellation is completed, at least in Vibrio species.

That's a valid observation. We acknowledge that FlhF can indeed be observed at the flagellated cell pole. We would hypothesize that FlhF might no longer be part of a mature flagellum. Whether it remains associated with HubP (or other factors such as FipA) is also a topic for further investigation. Regarding why FlhF and FlhG are omitted from panel D, it's due to the simplification of the model presented. If the reviewer concurs, we would prefer to maintain the current model to avoid unnecessary complexity.

5) Would you explain to me how does FlhG control flagellar number negatively? Does the negative regulation by FlhG not occur in Shewanella? In the proposed model, FlhG is necessary to assemble the C-ring via recruiting FliM/N to the HubP/FlhF-anchored FliF/G complex. Since deletion of FlhG increases the number of flagella at cell pole in Vibrio species, I am somewhat confused. This question is beyond the scope of this manuscript, but I am just curious to know the authors opinion.

Thank you for raising this relevant point. We referred to our previous work (i.e., Blagotinsek et al, 2020, PNAS) and updated discussion and model (Fig. 6). It now reads: "...Our previous findings in which FlhG connects flagellar C-ring assembly with the activity of the master transcriptional regulator FlrA19 align with the proposed model (Fig. 6e). FlrA and FliM share a binding site on FlhG, with their interaction dependent on FlhG's ATP-dependent dimerization state; FliM binds independently of nucleotide binding, whereas FlrA exclusively interacts with the ATP-bound FlhG dimer, stimulating its ATPase activity¹⁹. FlhG's recruitment to the cell pole is contingent upon the advancement of C-ring assembly, thereby affecting its interactions with both the master transcriptional regulator FlrA and the GTPase FlhF. ..."

Minor point

6) Introduction, line 22. It would be good to have one more reference that describes stimulation of FlhF GTPase by FlhG: Gulbranson et al, Mol. Microbiol. 2016.

True. Thank you. Updated.

7) Fig. 1b, legend: I assume this is FlhF of Shewanella putrefaciens, and source of protein should be noted.

Done.

8) Fig. 1c and in the text. For the general readers of Nature Communications, it would be good to clearly stated how different between "sub"-polar location and "lateral"-location.

Thank you. We have indicated the subpolar position of the flagellum in Fig. 1c, updated the figure legend accordingly, and refer to Fig. 1c in the text, when necessary.

9) Fig. 2b, lower left panel. In page 3, line 27; I wonder, which constructs did authors used for the X-ray crystal structure analysis. Did authors use pET24d_1-60N-Strep or pET24d_FlhF_1-137_N-Strep? I raised this question is because, I wonder whether following residues C-terminal to the residue 46 is non-structured (disordered) in the crystal or the construct does not contain those residues.

Good point. We employed a His-tagged construct for crystallization, which contained the first 44 amino acid residues of the B-domain (see Supplementary Table; pET24d_FlhF_1-44_C-His). The "60" was a typo, and was supposed to be "44", as described in the plasmid table. We revised the text accordingly.

10) Fig. 4a, lower panels. For me, it is a little bit difficult to see the bipolar localization of FlhF-mVenus. For example, the fluorescent dot at the bottom of the rightmost cell in FliGΔN/FlhF-mVenus image is so faint, even though it is highlighted by the arrowhead.

The reviewer's assessment is accurate; indeed, some of the focal points were quite faint. To enhance their visibility, we have adjusted the strength of the cell outlines in Figure 4a. Moreover, to provide further clarity, we have included an additional panel at the bottom of Figure 4a. This new panel offers enlarged views of cells exhibiting subtle signals in each sample. The figure legend was updated accordingly.

11) page 5, line 47; Did author not detect any fluorescence signal from the cells expressing mVenus-FliG alone? If so, it would be good to state that facts.

We apologize for any confusion caused by our previous statement. To clarify, when mVenus-FliG was expressed alone, no detectable fluorescence signal was observed.

12) In Fig. 5a, would it be possible to distinguish between the subpolar foci and bipolar foci by the color or symbol? It may be good for the general readers.

Thank you for the suggestion. We explored different versions, but unfortunately, all attempts led to images that were more distracting. As a result, we have chosen to retain the original figures.

13) Discussion, line 13, "Vibrio" should be italic.

Thanks. Done.

14) Discussion, line 38, I do not catch why authors assume "reasonable" for the polar insertion of FliF. In the previous phrase, it was stated that SecYEG machine is uniformly distributed.

OK. It now reads: "...However, and given the uniform distribution of SecYEG machines along the cytoplasmic membrane 34, insertion of FliF into the membrane can occur in close proximity to the pole. ..."

15) Fig. 1, legend, line 22; proteinS

Thank you. Corrected.

16) no page numbers were provided for reference 7, 10 and 27.

Fixed.

Reviewer #2 (Remarks to the Author):

FliH and FliG control the location and number of flagella in many polar flagellated bacteria, although the mechanisms by which these proteins affect the flagellation pattern are poorly understood. The authors show here that the GTPase (NG) domain of FliH interacts with the polar landmark protein HubP and a region within the N-terminal B-domain of FliH (residues 1-46; FliG interaction domain, FID) binds FliG in *Shewanella putrefaciens*. FliH binds specifically to the FliG associated with polar flagella as it does not bind to the FliG paralog that forms lateral flagella. The authors also show that FliH binds FliG and HubP simultaneously. They propose a model where the FliH-FliG complex binds HubP at the cell pole and FliF is then recruited to the pole to facilitate formation of the MS-ring. The data presented in the manuscript provide new information on addressing the mechanism by which FliH assists in targeting flagella to the cell pole, at least for bacterial species that have a HubP homolog.

Specific comments:

1. HubP/FimV homologs are relatively limited in their phylogenetic distribution, which raises the question how applicable the proposed model is for polar flagellated bacteria that lack HubP homologs.

*This reviewer's point is crucial. There are polar flagellated species, such as *Caulobacter*, which exhibit polar flagella but lack HubP, FliH, and FliG. Therefore, our model likely applies to polar flagellates that establish their flagella via FliH/FliG involving HubP. Additionally, besides HubP, other factors like the recently identified FipA seem to contribute to establishing polar flagellation. To address this significant question raised by the reviewer, we have expanded our discussion accordingly.*

It would enhance the impact of the manuscript if the authors could demonstrate that a FliH protein from bacterial species that lacks a HubP homolog also binds FliG.

A notable species lacking HubP is Bacillus subtilis, which establishes its peritrichous flagellation pattern via FlhF and FlhG. Although not within the scope of this manuscript, we would like to confidentially inform the referee that in B. subtilis, FlhF does not bind to FliG due to a distinct FID domain. Instead, our experiments indicate that the FID domain of B. subtilis FlhF directly interacts with the C-terminal domain of FliF. However, these experiments, along with the differences in the interaction networks of FlhF (and FlhG, as also discussed in Schuhmacher et al. 2015, PNAS), are currently undergoing further investigation and will be addressed in another manuscript.

2. It would be informative if the authors included data on the localization of HubP in *S. putrefaciens*. Does HubP localize to both cell poles in *S. putrefaciens* as the authors show it does in *Escherichia coli* in Figure 5a? If HubP localizes to both cell poles in *S. putrefaciens*, some discussion of what might prevent the FlhF-FliG complex from binding to HubP at both cell poles would be helpful.

Thank you for bringing up this issue. We have included information regarding HubP localization in S. putrefaciens in the third part of the Results section. In summary, HubP forms a prominent cluster at the flagellated pole and a smaller cluster at the opposite pole. However, the mechanism by which polar identity is established remains unknown — it's a fascinating question that still awaits elucidation.

3. It is not clear how the observation that the FlhF-mVenus protein localizes to both cell poles when the N-terminus or C-terminus of FliG is deleted fits into the model shown in Figure 6 since the FlhF Δ N44 variant localizes exclusively to one cell pole. Given that the FlhF variant does not bind FliG, it is not obvious how the truncated FliG proteins affect the localization of the FlhF Δ N44 variant.

It appears that interactions with incomplete FliG (lacking either its N- or C-terminus) result in bipolar localization. A compelling hypothesis is that this complex loses its proper pole designation. We have incorporated this into the second paragraph of the Discussion section.

Minor comments:

1. Update reference 17 and change the name FIP to FipA to reflect the updated reference.

OK.

2. Additional details in legend to Figure 1 would be helpful. For example, explain the HIS3 and ADE2 reporters.

We have now mentioned in the legend to Figure 1 that the shown interaction analysis is a yeast two-hybrid (Y2H) assay. More details about the used Y2H system and how the interaction strength is scored can be found in the Materials and Methods section; moreover, the used Y2H plasmids are listed in Supplementary Table 4. We updated the figure legend with additional details with respect to reporter strain, reporter genes and promoters.

Also, why does the colony color vary among the strains and growth media?

Depending on the strength of the Y2H interaction, different amounts of these two enzymes will be produced from the HIS3 and ADE2 reporter genes; thereby, enabling different extents of growth of PJ69-4A cells on SC-HLT (lacking histidine) or SC-ALT (lacking adenine) plates. As the SC-LT and SC-HLT plates contain only half of the standard adenine amount (10 mg/l instead of 20 mg/l adenine hemisulfate), cells not producing additional Ade2 enzyme from the GAL2-ADE2 reporter show normal growth on SC-LT plates (selection for the two Y2H plasmids), but exhibit a red colony colour. The reason for the red colony colour is the vacuolar accumulation of a red pigment, which is generated from 5'-phosphoribosyl-5-aminoimidazole (AIR) when this adenine precursor cannot be further metabolized to 5'-phosphoribosyl-4-carboxy-5-aminoimidazole (CAIR) due to the absence or insufficient levels of the Ade2 enzyme. Depending on the levels of the produced Ade2 enzyme and, hence, on the efficiency of AIR to CAIR conversion, cells will accumulate the red, AIR-derived pigment to different extents and display colony colours ranging from red, light red, pink to white; with white being the normal colony colour of wild-type yeast cells (i.e., containing the wild-type ADE2 gene under the transcriptional control of the ADE2 promoter). Accordingly, only a strong Y2H interaction results in good growth both on SC-HLT and SC-ALT plates and white colony colour on all three media. In the absence of a Y2H interaction, cells will only grow on SC-LT plates and display a red colony colour. Interactions of intermediate strength will lead to pink colony colour, mostly normal growth on SC-HLT plates and slower growth on SC-ALT plates. Weak interactions will lead to light red to red colony colour on SC-LT plates, slower growth on SC-HLT plates and strongly reduced or no growth on SC-ALT plates. The used Y2H system therefore allows to obtain a rough estimate of the interaction strength between two proteins.